# Unravelling Phytochemical and Bioactive Potential of Three *Hypericum* Species from Romanian Spontaneous Flora: *H. alpigenum*, *H.* *perforatum* and *H. rochelii*

**DOI:** 10.3390/plants11202773

**Published:** 2022-10-19

**Authors:** Mihai Babotă, Oleg Frumuzachi, Andrei Mocan, Mircea Tămaș, Maria Inês Dias, José Pinela, Dejan Stojković, Marina Soković, Alexandru Sabin Bădărău, Gianina Crișan, Lillian Barros, Ramona Păltinean

**Affiliations:** 1Department of Pharmaceutical Botany, “Iuliu Hațieganu” University of Medicine and Pharmacy, Gheorghe Marinescu Street 23, 400337 Cluj-Napoca, Romania; 2Laboratory of Chromatography, Institute of Advanced Horticulture Research of Transylvania, University of Agricultural Sciences and Veterinary Medicine, 400372 Cluj-Napoca, Romania; 3Centro de Investigação de Montanha (CIMO), Instituto Politécnico de Bragança, Campus de Santa Apolónia, 5300-253 Bragança, Portugal; 4Laboratório Associado para a Sustentabilidade e Tecnologia em Regiões de Montanha (SusTEC), Instituto Politécnico de Bragança, Campus de Santa Apolónia, 5300-253 Bragança, Portugal; 5Institute for Biological Research “Siniša Stanković”-National Institute of Republic of Serbia, University of Belgrade, Bulevar despota Stefana 142, 11060 Belgrade, Serbia; 6Faculty of Environmental Sciences and Engineering, Babeș-Bolyai University, 30, Fântânele Street, 400294 Cluj-Napoca, Romania

**Keywords:** *Hypericum* species, *Hypericum rochelii*, *Hypericum alpigenum*, antioxidant activity, enzyme inhibition

## Abstract

*Hypericum perforatum* L., also known as St. John’s Wort, is recognized worldwide as a valuable medicinal herb; however, other *Hypericum* species were intensively studied for their bioactive potential. To fill the research gap that exists in the scientific literature, a comparative evaluation between *H. alpigenum* Kit., *H.* *perforatum* L. and *H. rochelii* Griseb. & Schenk was conducted in the present study. Two types of herbal preparations obtained from the aerial parts of these species were analyzed: extracts obtained through maceration and extracts obtained through magnetic-stirring-assisted extraction. LC-DAD-ESI-MS^n^ analysis revealed the presence of phenolic acids, flavan-3-ols and flavonoid derivatives as the main constituents of the above-mentioned species. Moreover, all extracts were tested for their antioxidant, enzyme-inhibitory and antimicrobial potential. Our work emphasizes for the first time a detailed description of *H. rochelii* phenolic fractions, including their phytochemical and bioactive characterization. In comparison with the other two studied species, *H. rochelii* was found as a rich source of phenolic acids and myricetin derivatives, showing important antioxidant, anticholinesterase and antibacterial activity. The study offers new perspectives regarding the chemical and bioactive profile of the less-studied species *H. alpigenum* and *H. rochelii*.

## 1. Introduction

The *Hypericum* genus comprises more than 500 species, which are commonly found in temperate regions both as spontaneous or cultivated plants [1]. Among them, *Hypericum perforatum* L. (St. John’s Wort) is the most well-known one, especially for its applications in human medicine, which are related to various phytochemicals contained in the aerial parts of this species. Naphtodianthrones (i.e., hypericins), the main interest constituents of this herb, were intensively studied for their pharmacological applications, their effectiveness in treatment of mild to moderate depressive episodes being proven by multiple clinical trials and also by well-established long-term use of the herbal drug in traditional medicine [2]. Moreover, hypericins were cited as valuable natural biomolecules with antiproliferative effects [3,4,5]. The therapeutic relevance of *H. perforatum* is also supported through the presence of other secondary metabolites, mainly flavonoids, tannins, xanthone and procyanidin derivatives, their occurrence being linked to use of different preparations obtained from this species as remedies for gastrointestinal, hepatobiliary or skin disorders [6,7,8].

Considering the multiple health benefits of St. John’s Wort, a high number of studies have evaluated other *Hypericum* species as potential candidates for therapeutic purposes. Even though, in recent decades, the interest in this topic has increased, there is still a lack of comprehensive knowledge regarding phytochemical and bioactive profiles of some members of this genera [9,10,11]. *Hypericum rochelii* Griseb. & Schenk is a Balkan species that expands in the south-western Carpathians of Romania at low altitudes (500–1200 m) on calcareous rocks [12,13], being previously studied for its volatile oil content and antibacterial properties [14,15]; no other data about the chemical composition or bioactive properties of this taxa were recorded. In a similar way, *Hypericum alpigenum* Waldst. et Kit. (sin. *Hypericum richeri* Vill. *ssp. grisebachii* (Vill.) Nym.) is a north Balkan–Carpathian element specific to the subalpine belt (1600–2000 m) [16,17]. Phenolic fractions and volatile oils [18,19] of this species were previously studied for their antioxidant [20], anti-inflammatory [21], antimicrobial [22] and gastroprotective effects [23].

In this context, the present study aimed to perform a comparative evaluation between the well-established *H. perforatum* L. and the less-studied *H. alpigenum* Kit. and *H. rochelii* Griseb. & Schenk, focused on the phytochemical and bioactive properties of those species. Hence, the phenolic profile of herbal preparations obtained from the above-mentioned taxa through classic maceration and magnetic-stirring-assisted (MSA) extraction was characterized using the LC-DAD-ESI-MS^n^ method. Additionally, the samples were tested for their *in vitro* antioxidant, enzyme-inhibitory and antibacterial properties.

## 2. Results

### 2.1. Phenolic Profile of the Extracts

The method employed for analysis of the phenolic profile of *H. perforatum*, *H. alpigenum* and *H. rochelii* allowed us to confirm the presence of 25 compounds (**Table 1**) that occur mainly in four structural categories: phenolic acid derivatives (**1**–**4**), flavan-3-ol derivatives (**5**–**8**, **10**, **11**, **14**), xanthone derivatives (**9**) and other flavonoids (**12**, **13**, **15**–**25**). Compounds **1**, **3** and **4** showed a common parent ion [M−H]^−^ at *m/z* 353, corresponding to caffeoylquinic acids, their identity being confirmed based on elution order and MS^2^ base peaks according to Clifford et al. [23,24]. Further, 3-*O*-caffeoylquinic acid (**1**) eluted first, showing the same MS^2^ base peak (*m/z* 191) with 5-*O*-caffeoylquinic acid (**4**), while 4-*O*-caffeoylquinic acid (**3**) was recognized after the MS^2^ deprotonated ion resulted after supplementary loss of water (*m/z* 191- 18 mu). Compound **2** ([M−H]^−^ at *m/z* 337) released the most abundant ion at *m/z* 162 (corresponding to coumaric acid) after MS^2^ fragmentation, being identified as 3-*O*-*p*-coumaroylquinic acid.

Flavan-3-ol derivatives (**5**–**8**, **10**, **11**, **14**) showed the same λ_max_ at 280 nm, which was previously reported as a distinctive feature for these compounds [25]. (+)-Catechin (**6**) and (-)-epicatechin (**7**) presented the same molecular ion [M−H]^−^ at *m/z* 289, their identity being assigned based both on their retention time, mass and UV–Vis characteristics and by comparison with available commercial standards. Compounds **5** and **14** showed a deprotonated ion [M−H]^−^ at *m/z* 577, releasing, after MS^2^ fragmentation, three main product ions: *m/z* 451 (−126 mu), 425 (−152 mu) and 407 (−152 to 18 mu); this fragmentation pattern was previously described as specific for B-type (epi)catechin dimers [26,27], allowing us to confirm their identity. In a similar way, the consecutive loss of fragments with *m/z* 288 ([epicatechin−H]^−^) from compounds **8, 10** and **11** indicated the presence of Β-type (epi)catechin trimer (**8**) and Β-type (epi)catechin tetramers (**10**, **11**).

Compound **9** (λ_max_ at 257 nm) showed a molecular ion [M−H]^−^ at *m/z* 421, which provided, after MS^2^ fragmentation, two major ions, namely *m/z* 331 and *m/z* 301; the fragmentation pattern was close to the one previously described for dibenzo-γ-pyrone-*C*-glycosides [28,29]. Hence, based on all these features, the compound was tentatively identified as mangiferin, the only xanthone derivative found in the analyzed species.

Compounds **13**, **15** and **19** were found as having the same molecular ion [M−H]^−^ at *m/z* 421, which released, after MS^2^ fragmentation, the most abundant fragment at *m/z* 317 ([myricetin−H]^−^) [30], their identities being assigned, respectively, as myricetin-*O*-hexoside (**13**) (loss of 162 mu) and myricetin-*O*-rutinoside (**15**, **19**). Quercetin derivatives (**12**, **16**–**18**, **20**, **22**, **26**) identities were assigned based on their λ_max_ around 354 nm and their distinct [M−H]^−^ at *m/z* 301 [31]. Compound **21** showed a molecular ion [M−H]^−^ at *m/z* 593, losing in MS^2^ a fragment of 308 mu, which corresponds to deprotonated rutinoside (rhamnosyl-hexoside; 146 mu + 162 mu), with a deprotonated ion at *m/z* 285; hence, the compound was tentatively identified as luteolin-*O*-rutinoside [32].

The quantitative distribution of the identified compounds in *H. perforatum*, *H. alpigenum* and *H. rochelii* extracts is summarized in **Table 1**. Among phenolic acid derivatives, 3-*O*-caffeoylquinic acid was the only one quantified in all extracts, reaching the highest concentrations in *H. alpigenum* (HAM, HAA). Further, 4-*O*-caffeoylquinic acid was found only in *H. alpigenum*, being more abundant for the extract obtained through MSA extraction; in a similar way, for the HRA extract, the most important content of 3-*O*-*p*-coumaroylquinic acid (98.09 ± 1.114 mg/g dry extract) was demonstrated. These results are in line with previous reports regarding the phenolic acids content of *H. perforatum* and *H. alpigenum* [25,33]. Epicatechin dimers (compounds **5** and **14**) were identified exclusively in *H. alpigenum*, both of them being found in high quantities in the extracts obtained through MSA extraction. Interestingly, myricetin derivatives could be found only in *H. rochelii* extracts, their quantitative distribution depending also on the extraction method; after phenolic acids, myricetin rhamosides were the main compounds of HRM and HRA. On the other hand, quercetin derivatives showed a preferential distribution in *H. perforatum* and *H. alpigenum*, quercetin-*O*-pentoside and quercetin-*O*-deoxyhexoside being the only ones that could be quantified in *H. rochelii* extracts. Mangiferin occured in all analyzed samples, the most important amounts of this compound being found in HAA and HRA. Xanthone derivatives, including mangiferin, were reported in other studies as valuable constituents of other *Hypericum* species [13,27,34]; even though they are usually found in small amounts, these secondary metabolites were suggested as potential markers for several *Hypericum* taxa, including *H. perforatum*, *H. capitatum* and *H. androsaemum* [20,35].

**Table 1 plants-11-02773-t001:** Chromatographic features and quantitative distribution of phenolic compounds identified through LC-DAD-ESI/MS^n^ in *Hypericum* extracts.

Peak No.	Rt (min)	λ_max_ (nm)	[M−H]^−^ (*m/z*)	MS^2^ (*m/z*)	Tentative Identification	Quantification (mg/g Extract)						Ref.
						HAA	HAM	HPA	HPM	HRA	HRM	
**1**	4.37	324, 298	353	191(100), 179(45), 135(10), 161 (5)	3-*O*-Caffeoylquinic acid ^1^	68.1 ± 1.192 ^bB^	30.34 ± 0.886 ^aC^	12.78 ± 0.097 ^aA^	14.11 ± 0.361 ^bB^	12.43 ± 0.113 ^bA^	7.45 ± 0.196 ^aA^	[25]
**2**	5.59	215, 31	337	163(100), 173(50), 191(25), 119(9), 135 (5)	3-*O*-*p*-Coumaroylquinic acid ^2^	nd	nd	3.03 ± 0.002 ^aA^	4.32 ± 0.095 ^b^	98.09 ± 1.114 ^aB^	nd	[36]
**3**	5.68	220, 310	353	173(100), 179(42), 191(24), 135(3)	4-*O*-Caffeoylquinic acid ^1^	23.17 ± 0.612 ^b^	12.89 ± 0.249 ^a^	nd	nd	nd	nd	[25]
**4**	6.16	289, 322	353	191(100), 179(13), 161 (<5), 135(<5)	5-*O*-Caffeoylquinic acid ^1^	nd	nd	10.91 ± 0.037 ^a^	15.76 ± 0.653 ^bA^	nd	50.56 ± 2.477 ^aB^	[37]
**5**	6.51	280	577	425(100), 407(68), 451(24), 289(22), 287(12)	Β-type (epi)catechin dimer ^3^	19.35 ± 0.318 ^b^	9.13 ± 0.369 ^a^	nd	nd	nd	nd	[27]
**6**	8.36	280	289	245(100), 205(36), 203(12), 179(6), 125(3)	(+)-Catechin ^3^	21.22 ± 0.621 ^bB^	9.86 ± 0.296 ^aB^	nd	nd	5.92 ± 0.065 ^bA^	3.79 ± 0.129 ^aA^	[27]
**7**	8.36	280	289	245(100), 205(32), 179(15), 203(7)	(-)-Epicatechin ^3^	nd	nd	11.96 ± 0.345 ^aB^	19.09 ± 0.003 ^bB^	8.33 ± 0.162 ^bA^	3.82 ± 0.008 ^aA^	[38]
**8**	9.29	280	865	695(100), 577(48), 713(33), 575(26), 287	Β-type (epi)catechin trimer ^3^	nd	nd	9.27 ± 0.035 ^a^	13.79 ± 0.002 ^b^	nd	nd	[34]
**9**	13.37	257, 274, 315	421	301(100), 331(91), 259	Mangiferin ^4^	6.52 ± 0.097 ^bC^	3.14 ± 0.149 ^aA^	2.18 ± 0.065 ^aA^	3.86 ± 0.138 ^bB^	5.44 ± 0.01 ^bB^	3.3 ± 0.012 ^aA^	[38]
**10**	10.73	280	1153	865(100), 577(54), 713(20), 287	Β-type (epi)catechin tetramer ^3^	nd	5.62 ± 0.859 ^aA^	10.62 ± 0.205 ^a^	15.63 ± 0.326 ^b^B	nd	nd	[38]
**11**	12.99	280	1153	865(100), 577(54), 713(20), 287	Β-type (epi)catechin tetramer ^3^	26.34 ± 0.357 ^b^	11.83 ± 0.082 ^a^	nd	nd	nd	nd	[36]
**12**	13.88	216, 354	625	463(100), 301(23)	Quercetin-*O*-dihexoside ^4^	nd	nd	0.49 ± 0.001 ^a^	1.34 ± 0.059 ^b^	nd	nd	[39]
**13**	14.18	265, 353	479	317(100)	Myricetin-*O*-hexoside ^5^	nd	nd	nd	nd	7.36 ± 0.113 ^b^	4.73 ± 0.432 ^a^	[37]
**14**	15.89	280	577	425(100), 451(40), 407	Procyanidin B5 ^3^	14.42 ± 0.043 ^b^	7.52 ± 0.194 ^a^	nd	nd	nd	nd	[39]
**15**	16.16	268, 359	463	317(100)	Myricetin-*O*-rhamnoside ^5^	nd	nd	nd	nd	66.59 ± 0.36 ^b^	42.87 ± 0.691 ^a^	[40]
**16**	16.41	255, 355	609	301(100)	Quercetin-*O*-deoxyhexosyl-hexoside ^4^	1.81 ± 0.001 ^bA^	0.58 ± 0.001 ^aA^	7.29 ± 0.023 ^aB^	9.44 ± 0.035 ^bB^	nd	nd	[39]
**17**	17.08	355	463	301(100)	Quercetin-*O*-hexoside ^4^	65.37 ± 1.932 ^b^	13.54 ± 0.002 ^a^	13.75 ± 0.11 ^a^	nd	nd	nd	[40]
**18**	17.14	290, 354	477	301(100)	Quercetin-3-*O*-hexuronide ^4^	nd	nd	nd	18.49 ± 0.086 ^a^	nd	nd	[39]
**19**	17.57	351	463	316(100), 317(70)	Myricetin-*O*-rhamnoside ^5^	nd	nd	nd	nd	15.48 ± 0.413 ^b^	10.94 ± 0.292 ^a^	[39]
**20**	19.85	267, 354	463	301(100)	Quercetin-*O*-hexoside ^4^	3.98 ± 0.11 ^b^	0.75 ± 0.016 ^aA^	nd	1.82 ± 0.009 ^aB^	nd	nd	[41]
**21**	19.69	213, 266, 347	593	285(100)	Luteolin-*O*-rutinoside ^4^	nd	nd	nd	2.59 ± 0.007 ^a^	nd	nd	[42]
**22**	19.72	352	505	301(100), 463(55)	Quercetin-*O*-acetyl-hexoside ^4^	nd	nd	2.1 ± 0.007 ^a^	nd	nd	nd	[25]
**23**	20.06	353	433	301(100)	Quercetin-*O*-pentoside ^4^	nd	nd	nd	nd	2.77 ± 0.016 ^b^	2.32 ± 0.057 ^a^	[42]
**24**	20.94	357	447	301(100)	Quercetin-*O*-deoxyhexoside ^4^	2.55 ± 0.096 ^bB^	0.8 ± 0.002 ^aA^	2.04 ± 0.015 ^aA^	3.9 ± 0.072 ^bB^	12.19 ± 0.073 ^bC^	7.41 ± 0.121 ^aC^	[25]
**25**	30.3	354	301	179(100), 151(24)	Quercetin ^4^	nd	0.58 ± 0.001 ^aA^	nd	3.77 ± 0.01 ^aB^	Nd	nd	[25]
					**Total Phenolic Acids**	91.27 ± 1.804 ^bB^	43.23 ± 1.135 ^aB^	26.73 ± 0.063 ^aA^	34.18 ± 1.109 ^bA^	110.52 ± 1.227 ^bC^	58.01 ± 2.281 ^aC^	
					**Total Flavan-3-ols**	81.33 ± 0.616 ^bC^	25.47 ± 0.04 ^a^	31.84 ± 0.175 ^aB^	48.51 ± 0.328 ^b^	14.25 ± 0.226 ^bA^	7.61 ± 0.137 ^a^	
					**Total Xanthonoids**	6.52 ± 0.097 ^bC^	3.14 ± 0.149 ^aA^	0.49 ± 0.001 ^aA^	3.86 ± 0.138 ^bB^	5.44 ± 0.01 ^bB^	3.3 ± 0.012 ^aA^	
					**Total Flavonoids**	73.71 ± 2.138 ^bB^	16.25 ± 0.016 ^aA^	27.36 ± 0.001 ^aA^	41.35 ± 0.123 ^bB^	104.39 ± 0.029 ^bC^	68.28 ± 0.374 ^aC^	
					**Total Phenolic Compounds**	252.83 ± 0.853 ^bC^	175.78 ± 1.392 ^aC^	86.42 ± 0.238 ^aA^	127.9 ± 0.766 ^bA^	234.61 ± 0.962 ^bB^	137.2 ± 2.78 ^aB^	

nd—not detected. Standard calibration curves: ^1^—chlorogenic acid (*y* = 168823*x* − 161172; *R*^2^ = 0.9999, LOD = 0.20 µg/mL and LOQ = 0.68 µg/mL); ^2^— *p*-coumaric acid (*y* = 301950*x* + 6966.7, *R*^2^ = 0.9999, LOD = 0.68 μg/mL; LOQ = 1.61 μg/mL); ^3^—catequin (*y* = 84950*x* + 23200, *R*^2^ = 1, LOD = 0.44 μg/mL; LOQ = 1.33 μg/mL); ^4^—quercetin-3-*O*-glucoside (*y* = 34843*x* − 160173, *R*^2^= 0.9998, LOD = 0.21 μg/mL; LOQ = 0.71 μg/mL); ^5^—myricetin-3-*O*-glucoside (*y* = 23287*x* − 581708, *R*^2^= 0.9988, LOD = 0.23 μg/mL; LOQ = 0.78 μg/mL). Statistical differences were assessed either by one-way ANOVA, followed by Tukey’s HSD post hoc test (α = 0.05), or by Student’s *t*-test (α = 0.05); lower-case letters indicate significant differences among extraction methods within the same species, whereas upper-case letters indicate significant differences among species within the same extraction method. **HAA** (*H. alpigenum* MSA extraction), **HAM** (*H. alpigenum* maceration), **HPA** (*H. perforatum* MSA extraction), **HPM** (*H. perforatum* maceration), **HRA** (*H. rochelii* MSA extraction), **HRM** (*H. rochelii* maceration).

### 2.2. Total Phenolic (TPC) and Total Flavonoid Content (TFC)

The results obtained after the assessment of TPC and TFC for the herbal preparations of *H. perforatum*, *H. alpigenum* and *H. rochelii* are presented in **Table 2**. As can be observed, the extracts obtained through MSA extraction showed higher phenolic contents compared to those obtained through maceration. Conversely, TFC varied independent of extraction method, the highest value being observed for HPM extract (100.17 ± 1.27 mg RE/g). A comparison between species based on these parameters highlights the highest values of total phenolic and flavonoid contents for *H. alpigenum*.

### 2.3. Antioxidant Potential of the Extracts

The antioxidant potential of the extracts obtained from *H. perforatum*, *H. alpigenum* and *H. rochelii* was evaluated through five in vitro complementary methods, the results being presented in **Table 2**. The HPM and HRA extracts exerted the strongest free radical scavenger activity (TEAC assay), a similar trend being described for the same extracts regarding ferric ion reducing power (FRAP) and oxidative hemolysis inhibition (OxHLIA assay). Nevertheless, a clear interdependence between phenolic content and antioxidant activity could not be established, which indicates that other phytoconstituents contained in the samples made an important contribution to their antioxidant potential.

The high to moderate antioxidant potency of the analyzed extracts is supported also by the results obtained in TBARS and OxHLIA assays, which were proven as being more sensitive than the conventional antioxidant methods [43]. Usually, in the OxHLIA kinetic assay, IC_50_ values (µg/mL) are calculated for a given period of time by correlating the extract concentrations to the Δ*t* values (min) (calculated based on half hemolysis time obtained from the hemolytic curves of each extract concentration minus the H*t*_50_ value of the negative control). For a 60 min Δ*t*, all extracts presented lower IC_50_ values than the positive control (Trollox), the same trend being described for all samples in the TBARS assay.

### 2.4. Enzyme-Inhibitory Activity of the Extracts

All the extracts were tested for their anti-glucosidase, anti-tyrosinase and anti-cholinesterase potentials, the results being summarized in **Table 3**; additionally, inhibition curves for each active extract are presented in **Figure 1**. Overall, the analyzed species showed weak inhibitory activity on tyrosinase (HPA was the only one that exerted anti-tyrosinase activity), promising anti-glucosidase activity (all the extracts reached IC_50_ values at lower concentrations than the positive control) and moderate anti-cholinesterase potential. Several differences could be observed between species in terms of inhibitory potential, *H. perforatum* samples acting more as acetylcholinesterase and *α*-glucosidase inhibitors, while, for *H. alpigenum,* the most prominent *α*-glucosidase inhibition was observed (an IC_50_ value of 17.35 ± 4.29 µg/mL extract for HMA).

### 2.5. Antimicrobial Properties of the Extracts

In addition to well-established pharmacological properties (i.e., antidepressant, antioxidant, etc.), some *Hypericum* species were cited as possessing antibacterial and/or antifungal activity, proven by *in vitro* methods [22,44,45,46,47]. In this regard, we aimed to conduct a comparative evaluation of the antimicrobial potential of the extracts obtained from *H. perforatum*, *H. alpigenum* and *H. rochelii*, highlighting the main differences between this species in terms of antibacterial and antifungal effectiveness.

As can be observed in **Table 4**, all bacterial strains were sensible to the tested extracts, excluding HAM, which showed MIC and MBC values higher than 8 mg/mL. The most prominent antibacterial activity was exerted by *H. perforatum*, the lowest MIC and MBC being obtained for Gram-negative bacteria; conversely, *H. rochelii* was found as being more active on Gram-positive strains. In terms of the effectiveness of antibacterial potential, the extracts obtained through MSA extraction were found as being more active than those obtained by maceration. A comparative overview between the sensibility of the tested strains upon action of *Hypericum* extracts reveals *Staphylococcus aureus* as the most sensible among them.

The parameters that describe the antifungal activity of *Hypericum* extracts are summarized in **Table 5**. Overall, the tested strains showed moderate (*A. fumigatus*, *A. niger*, *A. versicolor*, *T. viride*) or weak (*P. funiculosum*, *P. verrucosum var. cyclopium*) sensibility. A similar trend for antibacterial potential was observed, *H. perforatum* extracts showing the lowest MIC and MFC values, especially for the extract obtained through MSA extraction (HPA), which exerted good antifungal activity on *A. niger* and *A. versicolor* (MIC—1 mg/mL, MFC—2 mg/mL for both strains).

## 3. Discussion

After an overview of the LC-DAD-ESI/MS^n^ results, it could be suggested that magnetic-stirring-assisted (MSA) extraction induced higher recovery yields for the phenolic compounds contained in the analyzed samples, this trend being more visible in the case of *H. alpigenum* and *H. rochelii* extracts. Moreover, our findings showed that each extraction method needs to be customized for each type of main compound that will be extracted; in fact, MSA increased the recovery of phenolic acids, xanthonoids and flavonoids, while maceration was more effective in terms of flavan-3-ols extraction yields. In addition, our phytochemical assessment provides the first detailed report about the qualitative and quantitative distribution of phenolic compounds from *H. rochelii*. As could be observed, our study emphasized the presence of myricetin derivatives, *p*-coumaroyl and caffeoylquinic acids as the main constituents of the above-mentioned species.

It must be noted that TPC and TFC assays offer a general overview about the amounts of phenolic and flavonoid compounds contained in different samples and present some limitations regarding the interferences with other phytochemicals found in the analyzed matrices [48,49]. Nonetheless, they are still useful complementary tools in the chemical evaluation of plant extracts, the results obtained through these methods revealing correlations between antioxidant capacity of the samples or their chemical profile assessed by more sensible methods (i.e., liquid chromatography). Referring to the present study, the results obtained for TPC both in chromatographic and classic spectrophotometric evaluation indicate higher extraction yields for total phenolic compounds using MSA extraction. As well, the trend described for quantitative distribution of flavonoidic compounds by using TFC and LC–MS was the same for HAA, HPA, HPM and HRM extracts.

Initially, a clear interdependence between phenolic content and antioxidant activity could not be established, which indicated that other phytoconstituents of the samples made an important contribution to their antioxidant potential; hence, based on Pearson’s correlation coefficients, a correlogram was built in order to decipher the individual influence of each chemical constituent against the measured bioactivities, including the antioxidant one (**Figure 2**).

A strong positive correlation was found between TPC and oxidative hemolysis inhibition, while TFC was positively correlated with the ferric reducing power of the extracts. Regarding the influence of the individual phenolic constituents identified after LC-DAD-ESI/MS^n^ assessment, the most important contribution to the antioxidant activity of *Hypericum* samples could be attributed to the presence of 4-*O*-caffeoylquinic and 5-*O*-caffeoylquinic acids, as well as to the presence of (+)-catechin and procyanidin B5; the correlation coefficients obtained for the other identified constituents support the hypothesis that the antioxidant potential of the extracts could be influenced by some unidentified compounds, probably belonging to other chemical classes.

This aspect was previously highlighted for other *Hypericum* species; Radulović et al. showed that the antioxidant capacity of *H. perforatum* samples collected from the Balkans varied not exclusively depending on their phenolic content, revealing the contribution of several tannins to the total capacity of the extracts [45], while Gîtea et al. reported inconsistent variations in the antioxidant potential of several species rich in phenolic compounds collected from Romanian spontaneous flora (*H. perforatum* L., *H. maculatum* Cr *var*. *typicum* Frohlich., *H. hirsutum* L., *H. tetrapterum* Fr.) [50]. Moreover, the antioxidant potential of *H. rochelii*, was not reported yet by other studies. The extracts of this species strongly acted as free radical scavengers and lipid peroxidation inhibitors, showing also moderate capacity to act as reducing agents and oxidative hemolysis inhibitors. As could be observed, in comparison with *H. perforatum* and *H. alpigenum*, *H. rochelii* exhibited medium antioxidant capacity, the best results being described for the extracts obtained through MSA extraction.

Several studies were previously focused on evaluation of enzyme-inhibitory properties of *Hypericum* species [51,52,53,54]. Ethyl acetate, methanolic and aqueous extracts of *H. perforatum* L. were tested for their anti-cholinesterase and anti-tyrosinase activity by Altun et al., their findings indicating the highest acetylcholinesterase and low tyrosinase inhibition for the methanolic extract [51]. A moderate enzyme-inhibitory effect against acetylcholinesterase was also described for *H. olympicum*, *H. pruinatum* and *H. scabrum* collected from Turkey, while the same species showed important inhibition against *α*-glucosidase, correlated with the significant amounts of flavonoid derivatives quantified in their methanolic extracts [52]. To the best of our knowledge, there are no available data regarding the enzyme-inhibitory potential of *H. rochelii*, our study revealing for the first time the ability of the herbal preparations obtained from this species to act as *α*-glucosidase and acetylcholinesterase inhibitors. As can be observed, the anti-glucosidase effect was slightly enhanced in the case of the extract obtained through maceration, and the MSA extract showed better interaction with the acetylcholinesterase in terms of inhibitory activity. The correlation analysis revealed a strong interdependence between the anti-glucosidade activity of the extracts and their mangiferin and quercetin-*O*-deoxyhexoside content (positive correlation, *r* = 0.75 and *r* = 0.55, respectively), as can be observed in **Figure 2**. Even though the tyrosinase inhibition was weak, it could be positively correlated with the presence of quercetin-*O*-acetyl-hexoside in the HPA extract, while the anti-cholinesterase effects seem to be linked both with the highest total phenolic content of the extracts and several individual phenolic metabolites (especially caffeoylquinic acids and flavan-3-ols derivatives).

In terms of antimicrobial potence, *H. alpigenum* and *H. rochelii* were found as having comparable antifungal activity, little differences being observed only for *A. niger* and *T. viridae*, which were more sensible to the action of the extracts obtained from the second species. Đorđević et al. previously evaluated the antibacterial effect of *H. rochelii* against five bacterial (*Bacillus subtilis* ATCC 6633, *Staphylococcus aureus* ATCC 6538, *Escherichia coli* ATCC 8739, *Pseudomonas aeruginosa* ATCC 9027, *Salmonella abony* NCTC 6017) and two fungal strains (*Aspergillus niger* ATCC 16404 and *Candida albicans* ATCC 10231), revealing moderate activity of the volatile oil isolated from the aerial parts of this species against *B. subtilis*, *S. aureus* and *C. albicans*; no other studies indicated a antimicrobial effect of *H. rochelii* [15]. Hence, our findings provide new perspectives regarding potential use of the phenolic fraction obtained from the above-mentioned species as a mild antibacterial agent, encouraging further in-depth evaluation for additional mechanisms that could be involved in this activity (i.e., inhibition of biofilm or pyocyanin formation) [55,56].

## 4. Materials and Methods

### 4.1. Standards, Reagents, and Other Chemicals

The following reagents and standards were used for determination of total phenolic and flavonoid contents, and antioxidant assays: Folin–Ciocâlteu reagent, Na_2_CO_3_, AlCl_3_ × 7H_2_O, 2,2′-azino-bis(3-ethylbenzothiazoline-6-sulfonic acid) diammonium salt (ABTS), K_2_S_2_O_8_, CH_3_COONa, CH_3_COOH, HCl, FeCl_3_ × 6H_2_O, 2,4,6-tris(2-pyridyl)-S-triazine (TPTZ), 2,2-diphenyl-1-picrylhydrazyl (DPPH), trichloroacetic acid (TCA), 6-hydroxy-2,5,7,8-tetramethylchroman-2-carboxylic acid (Trolox) and 2,2′-azobis(2-methylpropionamidine) dihydrochloride (AAPH) (all from Sigma-Aldrich Chemie GmbH, Schnelldorf, Germany). For the enzyme-inhibitory assays, the following reagents and standard were used: phosphate-buffered saline (PBS), acetylcholinesterase from electric eel (C3389), *α*-glucosidase from *Saccharomyces cerevisiae* (G5003) and tyrosinase from mushroom (T3824); acarbose, galantamine, kojic acid, dimethyl sulfoxide (DMSO), 5,5′-dithiobis(2-nitrobenzoic acid) (DTNB), levodopa (L-DOPA), 4-nitrophenyl-*β*-D-glucopyranoside (*p*-NPG), Tris base (all from Sigma Aldrich Chemie GmbH, Schnelldorf, Germany). All other reagents were purchased from local suppliers.

For LC–MS analysis, HPLC grade ethyl acetate (≥99%), acetonitrile, methanol and acetic acid (≥99%) were purchased from Merck (Darmstadt, Germany). Gallic acid, catechin, chlorogenic acid (3-CGA), 4-hydroxybenzoic acid, vanillic acid, epicatechin, syringic acid, 3-hydroxybenzoic acid, 3-hydroxy-4-methoxybenzaldehyde, *p*-coumaric acid, rutin, sinapinic acid, *t*-ferulic acid, naringin, 2,3-dimethoxybenzoic acid, benzoic acid, *o*-coumaric acid, quercetin, harpagoside, *t*-cinnamic acid, naringenin and carvacrol (purity > 98%) purchased from Sigma-Aldrich (Milan, Italy) were used as standards for LC–MS analysis. Ultra-pure water was obtained using a Millipore Milli-Q Plus water treatment system (Millipore Bedford Corp., Bedford, MA, USA).

### 4.2. Plant Material

During plants’ maximum flowering period (when all the floral buds of the plant were completely opened), aerial parts of *Hypericum* species were collected from Romanian spontaneous flora as following: *Hypericum alpigenum* Kit.–Sun Valley, Godeanu Mountains (August 2014), *H. perforatum* L.–Ploscoș, Cluj County (August 2014) and *H. rochelii* Griseb. & Schenk–Iron Gates, Drobeta Turnu-Severin (August 2014). The raw plant material was sorted and authenticated through confirmation of morpho-anatomical features of each species (available in Flora Europea and Flora of Romania), subsequently being subjected to a drying procedure at room temperature until it reached a constant mass. Afterwards, dried plant material was kept in paper bags in the herbarium of Pharmaceutical Botany Department of “Iuliu Hațieganu” University of Medicine and Pharmacy Cluj-Napoca until extraction.

### 4.3. Extraction Procedure

For ensuring uniformity of the plant material used for extraction, it was powdered using a laboratory mill (Grindomix^®^ GM 200, Retsch Gmbh., Germany) and manually sieved (1 mm standard sieve according to PhEur 10.6). In order to achieve extraction of bioactive compounds, two parallel extraction methods were implemented: magnetic-stirring-assisted (MSA) extraction and conventional maceration. For MSA extraction, 5 g of each powdered plant material was mixed with ethanol 70% (1:10 *w/v* ratio), and the resulted mixture was subsequently placed on the magnetic stirrer for 15 min, at a temperature of 40 °C. Likewise, 5 g of each powdered plant material was mixed with ethanol 70% in a ratio of 1:10 (*w/v*), and the resulted mixture was subsequently placed in a dark place, at room temperature, for 10 days, for maceration to occur. After the extraction procedure, the extracts were filtered under reduced pressure, concentrated until complete evaporation of the alcohol (using a rotary evaporator), freeze-dried and stored in a desiccator, protected from light and at room temperature. Hence, 6 herbal preparations were obtained, namely: **HAA** (*H. alpigenum* MSA extraction), **HAM** (*H. alpigenum* maceration), **HPA** (*H. perforatum* MSA extraction), **HPM** (*H. perforatum* maceration), **HRA** (*H. rochelii* MSA extraction), **HRM** (*H. rochelii* maceration).

### 4.4. LC-DAD-ESI/MS^n^ Characterization of Phenolic Profile

Ten milligrams of each dry extract were redissolved in 2 mL of ethanol/water (20:80, *v/v*) and filtered through 0.22-*μ*m disposable LC filter disks before injection. An Dionex Ultimate 3000 UPLC (Thermo Scientific, San Jose, CA, USA) system equipped with a diode array detector coupled to an electrospray ionization mass detector (LC-DAD-ESI/MS^n^) was employed for analysis of phenolic compounds using a method previously described [39]. Chromatographic separation was conducted on a Spherisorb S3 ODS-2C18 column (3 μm, 4.6 mm × 150 mm, Waters, Milford, MA, USA) using as solvents 0.1% aqueous formic acid (A) and acetonitrile (B) in gradient elution: isocratic 15% B (5 min), 15% B to 20% B (5 min), 20–25% B (10 min), 25–35% B (10 min), 35–50% B (10 min) and re-equilibration of the column using a flow rate of 0.5 mL/min. Online detection was achieved using a Diode Array Detector DAD (280, 330 and 370 nm as preferential wavelengths) coupled with an ESI mass spectrometer working in negative mode (Linear Ion Trap LTQ XL mass spectrometer, Thermo Finnigan, San Jose, CA, USA). Identification of phenolic compounds was made by comparing their retention times and UV–Vis and mass spectra with those obtained from standard compounds (when available); otherwise, compounds were tentatively identified comparing the obtained information with available data reported in the literature. For the quantitative evaluation, a calibration curve for each available phenolic standard (Extrasynthèse, Genay, France) was constructed based on the UV signal; for the identified phenolic compounds for which a commercial standard was not available, the quantification was performed through the calibration curve of the most similar available standard, and results were expressed as mg/g of extract [57].

### 4.5. Evaluation of Total Phenolic (TPC) and Total Flavonoid (TFC) Contents

To evaluate total phenolic content (TPC) of the assessed species, Folin–Ciocalteu (F-C) method was implemented. In a 96-well plate, 100 μL of 10% F-C solution were mixed with 20 μL of sample solution and pre-incubated for 3 min at room temperature in a dark place. Subsequently, the mixture was completed with 80 μL of 7.5% Na_2_CO_3_ solution and the resulted mixture was incubated for another 30 min in the same conditions. Finally, the absorbance of the mixture was read at 760 nm, and the results were expressed as milligrams of gallic acid equivalents/g of lyophilized extract (mg gallic acid equivalents − GAE/g extract) [43].

Conversely, to evaluate total flavonoid content (TFC) of assessed plants, in a 96-well plate, 100 μL of 2% AlCl_3_ solution were mixed with 100 μL of sample solution and incubated for 10 min at room temperature in a place free of light. The absorbance of the mixture was read at 420 nm, and the results were expressed as milligrams of rutin equivalents/g of lyophilized extract (mg rutin equivalents − RE/g extract) [43].

### 4.6. Total Antioxidant Capacity

Total antioxidant capacity of the extracts was measured using five different complementary assays, namely: Trolox Equivalent Antioxidant Capacity (TEAC), Ferric Reducing Antioxidant Power (FRAP), 2,2-diphenyl-1-picrylhydrazyl (DPPH) radical-scavenging activity, Thiobarbituric Acid Reactive Substances (TBARS) formation inhibition capacity and Oxidative Hemolysis Inhibition Assay (OxHLIA).

#### 4.6.1. TEAC Assay

To generate a radical stock solution, 50 mL of ABTS^+^ (2.13 mM) were mixed with 50 mL of K_2_S_2_O_8_ (1.38 mM), both reagents being dissolved in ultrapure water, which, after an incubation time of ~16 h in a dark place and at 20 to 25 °C, was subsequently diluted with distilled water to reach a final absorbance of the radical stock solution of 0.70 ± 0.02 at 734 nm. Afterwards, in a 96-well plate, 220 μL of reaction mixture, consisting of 200 μL of radical stock solution and 20 μL of sample solution, were incubated at room temperature in a dark place. After 6 min, the absorbance of the reaction mixture was read at 734 nm. The TEAC radical scavenging activity of the extracts was expressed as milligrams of Trolox equivalents/g of lyophilized extract (mg TE/g extract) [57].

#### 4.6.2. FRAP Assay

A FRAP reagent was generated by mixing 50 mL of acetate buffer (0.3 M, pH 3.6), 5 mL of FeCl_3_ solution (20 mM) and 5 mL TPTZ solution (10 mM) (both FeCl_3_ and TPTZ were dissolved in 40 mM HCl). Afterwards, in a 96-well plate, 200 μL of reaction mixture, consisting of 175 μL of FRAP reagent and 25 μL of sample solution, was incubated for 30 min in a dark place at room temperature; the final absorbance of the mixture was read at 593 nm. The FRAP radical scavenging activity of the extracts was expressed as milligrams of Trolox equivalents/g of lyophilized extract (mg TE/g extract) [43].

#### 4.6.3. DPPH Assay

A 0.004% DPPH radical solution was initially prepared by dissolving 5 mg of DPPH in 125 mL of methanol. Afterwards, in a 96-well plate, 270 μL of DPPH radical solution were mixed with 30 μL of sample solution. Following 30 min incubation period in a dark place and at room temperature, the absorbance of the reaction mixture was read at 517 nm, and DPPH radical scavenging activity of the extracts was expressed as milligrams of Trolox equivalents/g of lyophilized extract (mg TE/g extract) [57].

#### 4.6.4. TBARS Assay

In a pre-incubation phase, 200 μL of sample solution (extracts of each sample were serially diluted) was mixed with 100 μL of FeSO_4_ (10 μM) and 100 μL of ascorbic acid (0.1 mM) in an Eppendorf reaction tube (2 mL). The mixture was pre-incubated for 1 h at 37 °C. Afterwards, the reaction mixture was completed with 500 μL of trichloroacetic acid (28% *w/v*) and 380 μL of thiobarbituric acid (TBA, 2% *w/v*). The new resulted mixture was heated for 20 min at 80 °C. Finally, the reaction tubes were centrifuged at 3000× *g* for 10 min, and, in order to quantify malondialdehyde (MDA)-TBA complex, the absorbance of supernatant was read at 532 nm and results were expressed as EC_50_ values (μg/mL) [43].

#### 4.6.5. OxHLIA Assay

An erythrocyte solution (2.8%, *v/v*; 200 µL) prepared in phosphate-buffered saline (PBS, pH 7.4) was mixed with 400 µL of: either extract solution (2.03–130 µg/mL in PBS), Trolox (positive control; 7.81–125 µg/mL in PBS), PBS (negative control) or distilled water (baseline). The mixtures were pre-incubated for 10 min at 37 °C while continuously shaking. Afterwards, 200 μL of 2,2′-azobis(2-methylpropionamidine) dihydrochloride (AAPH; 160 mM in PBS) were added, and the optical density was kinetically measured at 690 nm in an ELx800 microplate reader (Bio-Tek Instruments, Winooski, VT, USA) until complete hemolysis. IC_50_ values (µg/mL) for a *Δt* of 60 min were obtained by correlating the extract concentration to the *Δt* values (min), which resulted from the half hemolysis time (H*t*_50_ values) obtained from the hemolytic curves of each extract concentration minus the H*_t50_* value of the PBS control [43].

### 4.7. Enzyme-Inhibitory Activity

Evaluation of enzyme-inhibitory activity of *H. alpigenum*, *H. perforatum* and *H. rochelii* included screening of these plants against *α*-glucosidase, tyrosinase and acetylcholinesterase using in vitro protocols.

### 4.7.1. α-Glucosidase Inhibition Assay

Using a protocol adapted for microplate reader, in a 96-well plate, a reaction mixture consisting of 50 μL of extract solution of different concentrations was mixed with 50 μL of *α*-glucosidase enzyme solution (0.75 U/mL), and 50 μL potassium phosphate buffer (100 mM, pH = 6.8) was pre-incubated for 10 min at 37 °C. Afterwards, 50 μL of *p*-NPG were added to the reaction mixture and plate was incubated for another 10 min at 37 °C. Finally, the absorbance was read at 405 nm and results were expressed as IC_50_ value (μg/mL). Acarbose was used as a positive control [43].

#### 4.7.2. Tyrosinase Inhibition Assay

Analogous to *α*-glucosidase inhibition assay, the protocol used for evaluating tyrosinase inhibition activity of the extracts was adapted for microplate reader. Therefore, 40 μL of different concentration of sample solutions were mixed with 80 μL of potassium phosphate buffer (50 mM, pH = 6.5) and 40 μL tyrosinase enzyme solution (125 U/mL) in a 96-well plate. The resulted mixture was pre-incubated for 5 min at 37 °C. Subsequently, 40 µL of L-DOPA (10 mM) were added and the new resulted mixture was incubated for another 15 min. Finally, the final absorbance of the reaction mixture was read at 492 nm, and results were expressed as IC_50_ value (μg/mL). Kojic acid was used as a positive control [57].

#### 4.7.3. Acetylcholinesterase Inhibition Assay

In a similar way as *α*-glucosidase and tyrosinase inhibition assays, a protocol adapted for microplate reader was used to evaluate acetylcholinesterase inhibition capacity of the extracts. Thus, 25 μL of different concentration of sample solutions were mixed with 50 μL of Tris-HCl buffer (50 mM, pH = 8), 125 μL of DTNB (0.9 mM) and 25 μL of acetylcholinesterase enzyme solution (0.078 U/mL). The reaction mixture was pre-incubated for 15 min at 37 °C. Subsequently, 25 μL of ATCI (4.5 mM) were added to the reaction mixture, and the plate was incubated for another 10 min at 37 °C. Finally, the absorbance was read at 405 nm and results were expressed as IC_50_ value (μg/mL). Galantamine was used as a positive control at varying concentrations [57,58].

### 4.8. Antimicrobial Activity

The extracts were tested for their antibacterial potential against Gram-positive bacteria *Staphylococcus aureus* (ATCC 11632), *Bacillus cereus* (clinical isolate), *Listeria monocytogenes* (NCTC 7973) as well as the following Gram-negative bacteria: *Escherichia coli* (ATCC 25922), *Salmonella enterica* subsp. *enterica* serovar. Typhimurium (ATCC 13311) and *Enterobacter cloacae* (ATCC 35030). For the antifungal assays, six micromycetes were used, namely *Aspergillus fumigatus* (human isolate), *Aspergillus niger* (ATCC 6275), *Aspergillus versicolor* (ATCC11730), *Penicillium funiculosum* (ATCC 36839), *Trichoderma viride* (IAM 5061) and *Penicillium verrucosum var. cyclopium* (food isolate). All strains were obtained from the Mycological Laboratory, Department of Plant Physiology, Institute for Biological Research “Siniša Stanković”, University of Belgrade, Serbia. The microdilution method was used to evaluate the parameters that describe antimicrobial efficiency: minimum inhibitory concentration (MIC, required for microbial growth inhibition), bactericidal (MBC) and fungicidal concentrations (MFC); all results were expressed as mg/mL [59].

### 4.9. Statistical Analysis

For each species, all the assays were carried out in triplicate. Statistical analysis was performed using GraphPad Prism 9 program. Differences were significant at the level of *α* = 0.05 by using one-way analysis of variance (ANOVA) followed by Tukey’s HSD. To analyze the relationship between different outcome variables, correlation analysis was performed using RStudio software (RStudio Desktop 2022.07.2+576) [43,60]. All the data were expressed as mean values with standard deviations (mean ± SD).

## 5. Conclusions

We evaluated the phenolic and bioactive profile of the extracts obtained through maceration and magnetic-stirring-assisted extraction from the aerial parts of *H. perforatum, H. alpigenum* and *H. rochelii* collected from Romanian spontaneous flora. Even though the correlation analysis proved interdependence between the bioactive profile of this species (antioxidant and enzyme-inhibitory properties) and their phenolic profile (both total and individual phenolic contents), their therapeutic potential could be linked with the presence of the other chemical constituents found in the extracts. The originality of the present study consists in the first report about the phenolic composition and bioactive profile of *H. rochelii*, a less studied species belonging to the *Hypericum* genus. Herbal preparations obtained from *H. rochelii* were found as containing high amounts of phenolic acids and myricetin derivatives, exerting at the same time promising antioxidant and antibacterial activity, followed by moderate inhibitory potential against *a*-glucosidase and acetylcholinesterase. The obtained results encourage future in-depth evaluations on the chemical constituents of this species and the mechanisms involved in its bioactivities demonstrated in the present study.

## Figures and Tables

**Figure 1 plants-11-02773-f001:**
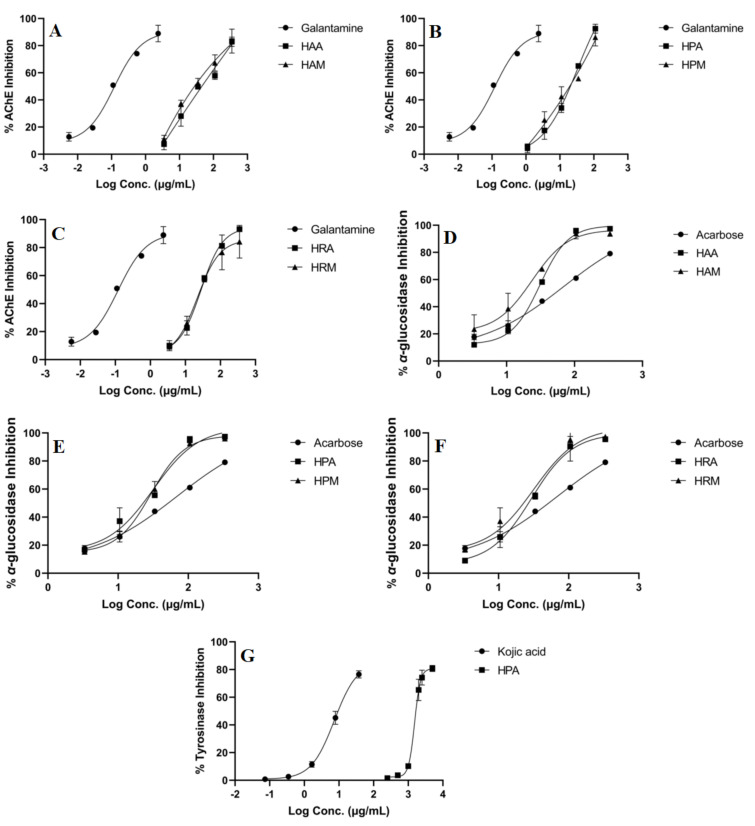
Enzyme inhibition curves of *Hypericum* extracts against acetylcholinesterase (AChE, (**A**–**C**)), *α*-glucosidase (**D**–**F**) and tyrosinase (**G**). **HAA** (*H. alpigenum* MSA extraction), **HAM** (*H. alpigenum* maceration), **HPA** (*H. perforatum* MSA extraction), **HPM** (*H. perforatum* maceration), **HRA** (*H. rochelii* MSA extraction), **HRM** (*H. rochelii* maceration).

**Figure 2 plants-11-02773-f002:**
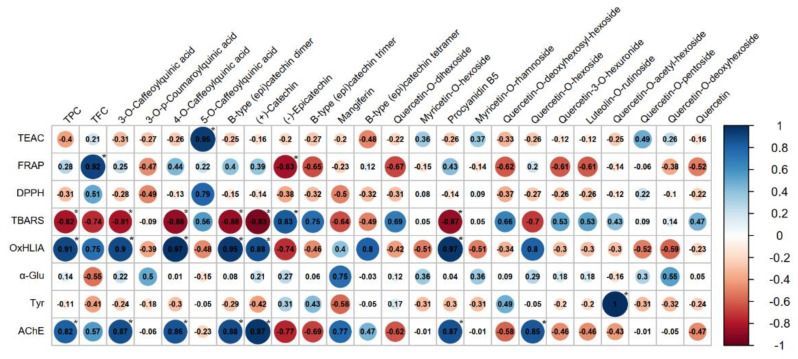
Correlation plot (correlogram) that shows Pearson’s correlation coefficients between chemical constituents of analyzed extracts and total antioxidant capacity and enzyme-inhibitory activities of the extracts. Red color represents negative correlation (values from 0 to −1), which indicates an inverse proportional relationship between the compared variables, while the blue color represents positive correlation (values from 0 to +1), which indicates a direct proportional relationship between the compared variables. The e diameter of each dot and the intensity of the colors indicate the strongness of correlation (increased values of Pearson’s correlation coefficients). Asterisk (*) indicates the statistical significant results (*p*<0.05).

**Table 2 plants-11-02773-t002:** Overview of TPC, TFC and antioxidant capacity values measured for *Hypericum* extracts.

Assay	HAA	HAM	HPA	HPM	HRA	HRM	Reference/ Control
**TPC** **(mg GAE/g)**	187.04 ± 6.64 ^bB^	159.66 ± 4.34 ^aB^	147.46 ± 5.09 ^aA^	140.44 ± 4.51 ^aA^	141.57 ± 3.63 ^aA^	134.55 ± 3.63 ^aA^	**Gallic acid** -
**TFC** **(mg RE/g)**	78.08 ± 1.32 ^aC^	100.17 ± 1.27 ^bC^	47.01 ± 0.86 ^bA^	39.42 ± 0.59 ^aA^	55.36 ± 1.39 ^aB^	75.95 ± 0.74 ^bB^	**Rutin** -
**TEAC** **(μg TE/g)**	257.83 ± 4.21 ^aA^	255.48 ± 4.41 ^aA^	253.93 ± 3.40 ^aA^	259.04 ± 4.19 ^aA^	253.70 ± 3.24 ^aA^	303.29 ± 4.76 ^bB^	**Trolox** 72.93 ± 0.56 *
**FRAP** **(μg TE/g)**	433.47 ± 2.04 ^aC^	524.93 ± 9.16 ^bC^	394.71 ± 13.44 ^bB^	316.96 ± 7.97 ^aA^	346.30 ± 8.52 ^aA^	487.26 ± 10.16 ^bB^	**Trolox** 120.85 ± 0.88 *
**DPPH** **(μg TE/g)**	196.75 ± 0.79 ^aB^	225.05 ± 4.31 ^bB^	204.84 ± 4.66 ^aB^	197.78 ± 3.79 ^aA^	187.19 ± 1.29 ^aA^	255.76 ± 4.69 ^bC^	**Trolox** 58.85 ± 0.48 *
**TBARS** **(μg/mL)**	3.05 ± 0.01 ^aA^	4.44 ± 0.04 ^bA^	9.71 ± 0.09 ^aC^	10.35 ± 0.03 ^bC^	6.36 ± 0.03 ^aB^	8.79 ± 0.01 ^bB^	**Trolox** 11.85 ± 0.03 *
**OxHLIA** **(μg/mL)**	21.40 ± 0.52 ^bB^*	19.30 ± 0.70 ^aB^	8.57 ± 0.36 ^aA^	8.46 ± 0.29 ^aA^	7.77 ± 0.20 ^aA^	8.57 ± 0.37 ^bA^	**Trolox** 21.72 ± 0.65 *

Statistical differences were assessed by one-way ANOVA, followed by Tukey’s HSD post hoc test (*α* = 0.05); lower-case letters indicate significant differences among extraction methods within the same species, whereas upper-case letters indicate significant differences among species within the same extraction method. The presence of asterisks (*) indicates statistical differences between reference compound and every extraction method according to Student’s *t*-test (*α* = 0.05). **HAA** (*H. alpigenum* MSA extraction), **HAM** (*H. alpigenum* maceration), **HPA** (*H. perforatum* MSA extraction), **HPM** (*H. perforatum* maceration), **HRA** (*H. rochelii* MSA extraction), **HRM** (*H. rochelii* maceration).

**Table 3 plants-11-02773-t003:** Overview of in vitro enzymatic inhibition potential of *Hypericum* extracts.

Enzyme	IC_50_ (μg/mL)						Positive Control
	HAA	HAM	HPA	HPM	HRA	HRM	
**α-Glucosidase** **(μg/mL)**	27.07 ± 0.82 ^bA^	17.35 ± 4.29 ^aA^	22.29 ± 4.62 ^aA^	24.88 ± 2.82 ^aA^	27.27 ± 4.12 ^aA^	22.29 ± 4.62 ^aA^	**Acarbose** 51.63 ± 2.40 *
**Tyrosinase** **(μg/mL)**	NA	NA	1664.49 ± 133.45	NA	NA	NA	**Kojic acid** 9.66 ± 1.70 *
**Acetylcholin-esterase** **(μg/mL)**	46.18 ± 9.60 ^bB^	28.66 ± 5.68 ^aA^	20.29 ± 0.99 ^aA^	19.63 ± 4.07 ^aA^	28.16 ± 1.52 ^aA^	29.05 ± 3.86 ^aA^	**Galantamine** 0.12 ± 0.001 *

Statistical differences were assessed by one-way ANOVA, followed by Tukey’s HSD post hoc test (*α* = 0.05); lower-case letters indicate significant differences among extraction methods within the same species, whereas upper-case letters indicate significant differences among species within the same extraction method. The presence of asterisks (*) indicates statistical differences between reference compound and every extraction method according to Student’s *t*-test (*α* = 0.05). **HAA** (*H. alpigenum* MSA extraction), **HAM** (*H. alpigenum* maceration), **HPA** (*H. perforatum* MSA extraction), **HPM** (*H. perforatum* maceration), **HRA** (*H. rochelii* MSA extraction), **HRM** (*H. rochelii* maceration).

**Table 4 plants-11-02773-t004:** Antimicrobial activity of *Hypericum* extracts. Streptomycin and ampicillin were used as control for bacterial growth. The results were given as minimum inhibitory concentration (MIC) and minimum bactericidal concentration (MBC) (all expressed as mg/mL).

Sample	MIC/ MBC	*Staphylococcus aureus*	*Bacillus cereus*	*Listeria monocytogenes*	*Escherichia coli*	*Salmonella* Typhimurium	*Enterobacter cloacae*
**HAA**	**MIC**	1	1	1	1	1	2
	**MBC**	2	2	2	2	2	4
**HAM**	**MIC**	>8	>8	>8	>8	>8	>8
	**MBC**	>8	>8	>8	>8	>8	>8
**HPA**	**MIC**	0.50	0.50	0.25	0.25	0.25	0.50
	**MBC**	1	1	0.50	0.50	0.50	1
**HPM**	**MIC**	0.25	1	1	0.50	0.50	1
	**MBC**	0.50	2	2	1	1	2
**HRA**	**MIC**	0.50	0.50	0.50	0.25	1	1
	**MBC**	1	1	1	0.5	2	2
**HRM**	**MIC**	0.25	1	1	1	0.50	1
	**MBC**	0.50	2	2	2	1	2
**Streptomycin**	**MIC**	0.10	0.025	0.015	1.0	0.10	0.025
	**MBC**	0.20	0.05	0.30	2	0.20	0.05
**Ampicillin**	**MIC**	0.10	0.10	0.15	0.5	0.15	0.10
	**MBC**	0.15	0.15	0.30	1	0.20	0.15

**HAA** (*H. alpigenum* MSA extraction), **HAM** (*H. alpigenum* maceration), **HPA** (*H. perforatum* MSA extraction), **HPM** (*H. perforatum* maceration), **HRA** (*H. rochelii* MSA extraction), **HRM** (*H. rochelii* maceration).

**Table 5 plants-11-02773-t005:** Antimicrobial and antifungal activities of *Hypericum* extracts. Ketoconazole and bifonazole were used as positive control for fungus growth. The results were given as minimum inhibitory concentration (MIC) and minimum fungicidal concentration (MFC) (all expressed as mg/mL).

Sample	MIC/ MFC	*Aspergillus fumigatus*	*Aspergillus niger*	*Aspergillus versicolor*	*Penicillium funiculosum*	*Penicillium verrucosum var. cyclopium*	*Trichoderma viride*
**HAA**	**MIC**	8	4	4	4	>8	2
	**MFC**	>8	8	8	8	>8	4
**HAM**	**MIC**	4	4	4	>8	4	>8
	**MFC**	8	8	8	>8	8	>8
**HRA**	**MIC**	4	2	4	4	8	2
	**MFC**	8	4	8	8	>8	4
**HRM**	**MIC**	4	8	4	>8	4	4
	**MFC**	8	>8	8	>8	8	8
**HPA**	**MIC**	2	1	1	4	2	4
	**MFC**	4	2	2	8	4	8
**HPM**	**MIC**	4	4	4	>8	>8	4
	**MFC**	8	8	8	>8	>8	8
**Ketoconazole**	**MIC**	0.15	0.15	0.10	0.2	0.10	0.10
	**MFC**	0.20	0.20	0.20	0.25	0.20	0.20
**Bifonazole**	**MIC**	0.20	0.20	0.20	0.20	0.20	1
	**MFC**	0.5	0.5	0.5	0.5	0.3	1.5

**HAA** (*H. alpigenum* MSA extraction), **HAM** (*H. alpigenum* maceration), **HPA** (*H. perforatum* MSA extraction), **HPM** (*H. perforatum* maceration), **HRA** (*H. rochelii* MSA extraction), **HRM** (*H. rochelii* maceration).

## Data Availability

Not applicable.

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
