# Peer review of "Unravelling Phytochemical and Bioactive Potential of Three *Hypericum* Species from Romanian Spontaneous Flora: *H. alpigenum*, *H.* *perforatum* and *H. rochelii"

_plants, 2022, doi:10.3390/plants11202773_

Round 1
Reviewer 1 Report
I appreciate that the paper brings the original results and broadens our knowledge of phytochemical composition and bioactive potential of less-studied Hypericum spp. However, it cannot be published in the presented form. Firstly, I would recommend to shorten the title (remove the abbreviations and replace "some" by "three").
ABSTRACT: The first sentences should be rephrased using appropriate and matter-of-act style.
INTRODUCTION: The very first paragraph must be rewritten. As the paper is not primarily focused on the evolutionary relations of Hypericum spp., this paragraph sounds generally and should be more concentrated on the phytochemical spectrum. The relation between studied species and subspecies mentioned in the lines 56-58 is not clear. Does it mean that the studied species are cryptic hybrids? Moreover, a great antiproliferative potential of hypericin is not mentioned at all.
RESULTS: The large amount of data provides a possibility to decipher potentially hidden relations. The representation and visualization of results could be improved using multidimensional statistical analysis.
It is not clear how the plant material was authenticated. Did the authors perform any cytogenetic analyses and/or DNA barcoding?
The results are somewhere shaded into discussion, especially in the first subchapter. The subsections of Results should not have the same titles like those in Material and methods.
Author Response
I appreciate that the paper brings the original results and broadens our knowledge of phytochemical composition and bioactive potential of less-studied Hypericum spp. However, it cannot be published in the presented form. Firstly, I would recommend to shorten the title (remove the abbreviations and replace "some" by "three").
We would to thank you for the appreciation showed for our work! As you recommended, we made several changhes in title in order to shorten it, including your suggestions.
ABSTRACT: The first sentences should be rephrased using appropriate and matter-of-act style.
Thank you! We rephrased the first part of the abstract.
INTRODUCTION: The very first paragraph must be rewritten. As the paper is not primarily focused on the evolutionary relations of Hypericum spp., this paragraph sounds generally and should be more concentrated on the phytochemical spectrum. The relation between studied species and subspecies mentioned in the lines 56-58 is not clear. Does it mean that the studied species are cryptic hybrids? Moreover, a great antiproliferative potential of hypericin is not mentioned at all.
Thank you for your valuable recommendation! As long as the other reviewers also made similar sugestions, the first part of the introduction was extensively modified, in order to emphasize more the state-of-the art regarding phytochemical and bioactive concerns regarding the studied species. We also added some information about antiproliferative effects of the main constituents of H. perforatum (hypericins).
RESULTS: The large amount of data provides a possibility to decipher potentially hidden relations. The representation and visualization of results could be improved using multidimensional statistical analysis.
Thank you! As long as one of the other reviewers also made a similar sugestion, we made a correlation analysis between the main chemical constituents of the extracts and their tested bioactivities, summarized in Figure 2.
It is not clear how the plant material was authenticated. Did the authors perform any cytogenetic analyses and/or DNA barcoding?
Authentication procedure was conducted by Prof. Dr. Alexandru Sabin Bădărău and Prof Dr Mircea Tămaș, based on the confirmation of morpho-anatomical features of each species available in dichotomic keys of Flora Europea and Flora of Romania. We updated this field in the text of the manuscript.
The results are somewhere shaded into discussion, especially in the first subchapter. The subsections of Results should not have the same titles like those in Material and methods.
Thank you! We introduced the requested changes in this section.

Reviewer 2 Report
Journal Name: Plants
Manuscript Number: plants-1933892
Title of the Manuscript: Unravelling Phytochemical and Bioactive Potential of some Hypericum Species from Romanian Spontaneous Flora: The Case of H. alpigenum Kit., H. perforatum L. and H. rochelii Griseb. & Schenk
Type of the Article: Original Research Article
REVIEWER’S COMMENT
Major revision comments
As a general observation, the study is extremely precise in the working methodology, analytical design and the presentation of results. The punctual comments, aside the analytical results obtained, also are accurate, and focused on the potential usage of low molecular weight fucosylated glycosaminoglycan sodium (LFG-Na) from the sea cucumber Holothuria fuscopunctata as a novel anticoagulant candidate.
1. Abstract
The novelty of this study should be stated more clearly.
2. Materials and methods:
Line 375-6: It was not clear from the paragraph regarding LC-DAD-MS method what is the previously described method, as there is not given a citation. Nevertheless, you should provide more details regarding the LC-MS method.
Lines 381-382: How you identified the components for which no standards were available? The phrase “were tentatively identified comparing the obtained information with available data reported in the literature” is not clear – if you used data from Database libraries you should specify what kind of database libraries you used and provide the web link to the database library.
Line 388-440: What kind of positive control you used for the determination of TPC , TFC, TEAC, FRAP, DPPH, TBRAS assays? How you performed the quantitative analyses?
3. Results and Discussion:
How you evaluated the relationship between phenolic content and antioxidant activity? If you used correlation or regression analysis, it should be specified.
The strengths and the limitations of the study should be clearly specified.
Author Response
Abstract
The novelty of this study should be stated more clearly.
Thank you for your valuable suggestion! We clearly emphasized the novelty of the study at the end of the section Abstract.
Materials and methods:
Line 375-6: It was not clear from the paragraph regarding LC-DAD-MS method what is the previously described method, as there is not given a citation. Nevertheless, you should provide more details regarding the LC-MS method.
Thank you! We updated the thext with the reference and some additionaly informations about the method.
Lines 381-382: How you identified the components for which no standards were available? The phrase “were tentatively identified comparing the obtained information with available data reported in the literature” is not clear – if you used data from Database libraries you should specify what kind of database libraries you used and provide the web link to the database library.
Thank you! The identity of each compound for wich standards were not available was confirmed by comparing its chromatographic features (retention time, maximum wavelenght, molecular ion m/z in negative mode and fragmentation pattern) with those already reported reported in other previously published articles (reference used for each compound were udated in were updated in Table 1, sectiun Results and discussion). No other Database libraries were used.
Line 388-440: What kind of positive control you used for the determination of TPC , TFC, TEAC, FRAP, DPPH, TBRAS assays? How you performed the quantitative analyses?
Thank you! TPC and TFC are quantitative methods used to appreciate total polyphenolic and flavonoid content of the extracts. Reference compounds used for calibration curves of these methods were gallic acid (for TPC; results expressed as mg gallic acid equivalents – GAE/g extract) and rutin (for TFC; results expressed as mg rutin equivalents – RE/g extract). For all the antioxindant activity assays, the positive control was trollox, results being expressed as trolox equivalents (μg TE/g extract). The names and values for each control were provided in results section (Table2), but we also mentioned them now in Materials and methods section.
Results and Discussion:
How you evaluated the relationship between phenolic content and antioxidant activity? If you used correlation or regression analysis, it should be specified.
Thank you! As long as one of the other reviewers also made a similar sugestion, we made a correlation analysis between the main chemical constituents of the extracts and their tested bioactivities, summarized in Figure 2
The strengths and the limitations of the study should be clearly specified.
Thank you! We emphasized this aspects by updating section Discussion.

Reviewer 3 Report
The authors have conducted an interesting study, where two types of extracts were prepared and assessed for their phytochemical, bioactive, and microbial potential of three species from the Hypericum genus. However, the manuscript requires extensive modifications before it can be considered fit to be published. I would highly recommend the authors for language editing. The introduction is not very well presented with some unnecessary details. I have added my comments and suggestions to the file attached below.

Author Response
The authors have conducted an interesting study, where two types of extracts were prepared and assessed for their phytochemical, bioactive, and microbial potential of three species from the Hypericum genus. However, the manuscript requires extensive modifications before it can be considered fit to be published. I would highly recommend the authors for language editing. The introduction is not very well presented with some unnecessary details. I have added my comments and suggestions to the file attached below.
L 4, 25: italics
Revised as requested.
Lines 45, 56-59
As long as the other reviewers also made similar sugestions, the first part of the introduction was extensively modified, in order to emphasize more the state-of-the art regarding phytochemical and bioactive concerns regarding the studied species. Please check the revised version of the manuscript.
L 74-75: A previous study also studied detailed composition of essential oil from this species and found 79 different compounds. Additionally, they also tested antimicrobial activity of the oils. Why this study was not included in the literature?
Thank you for this valuable recommendation! We updated this section.
L 75-80: Very long sentence, it is harder to read. Rewrite.
Revised as requested
L 89: superscript "n" is missing in the abstract. Make it uniform.
Revised as requested
L 96: Please follow mdpi citation rules
Revised as requested
L 127: At first glance, it appears as table title. Please avoid starting a paragraph like this.
The paragraph was revised
L 149: HPLC-DAD-ESI/MS or LC-DAD-ESI/MS ??
Title of the table was updated
Figure 1: Picture is of poor quality. Text are not very clear. Please update the picture.
We updated the picture.
Table 4: were statistics done here?
No statistics were not done at this point. When you test antimicrobial activity by serial dilution technique, under the same conditions and with the same concentrations, you always get the same results. So there is no SD and variation in values.
Table 4: heading missing
Heading was added
Table 4: why "," is used?
We replaced ”,” with ”.”
Table 4: Make the decimals uniform
Revised as requested
L 247: positive control
Revised as requested
Table 5: were statistics done here?
No statistics were not done at this point. When you test antimicrobial activity by serial dilution technique, under the same conditions and with the same concentrations, you always get the same results. So there is no SD and variation in values.
Table 5: heading missing
Heading was added
L 309: Please change to mdpi citation rules
Revised as requested
L 323: change the letter a in the word
Revised as requested
L 345: What do you mean by this?
Maximum flowering period is defined as the moment when all the the floral buds of the plant are completely opened. For Hypreicum species is considered the optimal harvesting time in order to assure the maximum content of bioactive compounds (especially hypericins).We updated this section in the text of manuscript.
L 347-348: Why the samples were not utilized right after the harvest or drying process? The samples harvested appears to be quite old. How do you think this time period might have effected the samples and the different analysis that were carried out in the this study?
Even that the samples were harvested a long time ago, the extraction procedure and all other analyzes were performed during 2017, the results being processed and prepared for publishing just at this time.
L 355: grammar
Revised as requested
L 356: grammar
Revised as requested
L 364: Duration of 10 days is quite long for maceration. Generally, it is between 3-7 days. Why 10 days were used?
Maceration was made following the pharmacopoeial recommendations listed in the monograph Tincturae (tinctures), published in Romanian Pharmacopoeia, 10th edition.
L 365: How much concentrated?
The extracts were concentrated in order to evaporate the alcohol before the freeze-drying procedure. We updated this information in the text of the manuscript.
L 365: Filtered using what?
The extracts were filtered using a vacuum filtration system. We updated this information in the text of the manuscript.
L 376: The citation should be added here
Thank you! As long as one of the other reviewers also made a similar sugestion, we updated the thext with the reference and some additionaly informations about the method.
L 390: Grammar (solution were mixed)
Revised as requested.
L 391-392: poor sentence formulation
The text was updated.
L 442: Italics. make it uniform across the manuscript
Revised as requested.
L 505-519: The whole conclusion can be rewritten in a better way. Sentences are hard to read, sentence formulation should be improved. Rewrite.
The Conclusions section was rewritten.

Round 2
Reviewer 1 Report
The manuscript should be published after several tiny corrections. Please see below.
Page 2, line 49 Please add "L.".
Page 2, lines 55-58; 82-85 The same sentences. Please chceck and correct the numbering of references.
Figure 2 Please improve the caption. Please clarify the color intensity and sizes of correlogram points.
Page 12, line 343 Please replace "an strong" by the correct article.
Page 12, line 345 Please mention if the correlation coeficients were statistically significant.
Page 17, line 558 Please replace "a correlation matrix was realized" by "correlation analysis was performed". Please cite RStudio IDE.
Author Response
The manuscript should be published after several tiny corrections. Please see below.
Thank you for your appreciation! We will improve the actual form of the manuscript by implementing your suggestions.
Page 2, line 49 Please add "L.".
Revised as requested.
Page 2, lines 55-58; 82-85 The same sentences. Please chceck and correct the numbering of references.
Revised as requested
Figure 2 Please improve the caption. Please clarify the color intensity and sizes of correlogram points.
We improved the caption of Figure 2; at the same time, we explained the significance of each element (diameter and colour of each point of correlogram).
Page 12, line 343 Please replace "an strong" by the correct article.
Revised as requested
Page 12, line 345 Please mention if the correlation coeficients were statistically significant.
Statistically significant results were marked in Figure 2 with an asterisk.
Page 17, line 558 Please replace "a correlation matrix was realized" by "correlation analysis was performed". Please cite RStudio IDE.
Revised as requested. Reference was added.
Reviewer 3 Report
The authors have worked on the manuscript and brought significant changes. I believe the changes done have improved the manuscript to an extent. However, the manuscript still needs to be worked on to refine it further. Some of the comments are mentioned below:
· The changes in the introduction section are not sufficient. Content-wise the introduction is still fine, although it lacks continuation and feels disconnected while reading (first paragraph). Additionally, a section of changes is repeated twice (lines 55-58)(lines82-85).
· The majority of the comments and suggestions were addressed. Although, some of the changes were ignored or not taken seriously by the authors. For example, the authors addressed a comment and changed the sentence (line 146), but the same thing was repeated in line 181.
· The updated figure 1 is slightly better than the last one but still appears pixelated.
· Make Typhimurium italics in Table 4.
· Line 25 change “fot” to “for”
· Line 48. please provide full botanical names of species when mentioned for the first time. (Make this change throughout the manuscript)
Author Response
The authors have worked on the manuscript and brought significant changes. I believe the changes done have improved the manuscript to an extent. However, the manuscript still needs to be worked on to refine it further. Some of the comments are mentioned below:
The changes in the introduction section are not sufficient. Content-wise the introduction is still fine, although it lacks continuation and feels disconnected while reading (first paragraph). Additionally, a section of changes is repeated twice
Thank you for your valuable recommendations! We introduced several changes in the Introduction section, including the deletion of the paragraph which was repeated twice.
The majority of the comments and suggestions were addressed. Although, some of the changes were ignored or not taken seriously by the authors. For example, the authors addressed a comment and changed the sentence (line 146), but the same thing was repeated in line 181.
Thank you! Unfortunately, at the first correction, we ommited to implement the requested change in line 181. We revised it now.
- The updated figure 1 is slightly better than the last one but still appears pixelated.
Probably, the quality of the picture decrease during the processing of the manuscript. We will provide it separately as attachement in order to be inserted further by the production office. At the same time , you can check again the quality of the picture in order to confirm if it is proper to be used.
- Make Typhimurium italics in Table 4.
The word `Typhimurium` remained written as non-italic as long as it is the official nomenclature recommendation for thys Salmonella serotype (https://doi.org/10.1128%2Fjcm.38.7.2465-2467.2000 ).
- Line 25 change “fot” to “for”
Revised as requested
- Line 48. please provide full botanical names of species when mentioned for the first time. (Make this change throughout the manuscript)
Revised as requested